# The Interplay of Sulfur and Selenium Enabling Variations in Micronutrient Accumulation in Red Spinach

**DOI:** 10.3390/ijms241612766

**Published:** 2023-08-14

**Authors:** Kashif Saeed, Fatiha Kalam Nisa, Muna Ali Abdalla, Karl Hermann Mühling

**Affiliations:** Institute of Plant Nutrition and Soil Science, Kiel University, Hermann-Rodewald-Str. 2, 24118 Kiel, Germany; ksaeed@plantnutrition.uni-kiel.de (K.S.); fknisa@plantnutrition.uni-kiel.de (F.K.N.)

**Keywords:** S, Se, micronutrients, molybdenum, zinc, manganese, iron, copper, total proteins, organic acids, water-soluble sugars

## Abstract

Aside from its importance in human and animal health, low levels of foliar-applied selenate (SeO_4_) can be advantageous in the presence of sulfur (S), contributing to improved growth, nutrient uptake, and crop quality. A hydroponic experiment in a growth chamber explored the interactive influence of Se and S on micronutrients and several quality indices, such as soluble sugars, organic acids, and total protein concentrations in spinach (*Spinacia oleracea* L.). Three levels of S (deprivation, adequate, and excessive) with varying quantities of Se (deficient, moderate, and higher) were examined in combination. Under S starvation and along with S nourishment in plant parts, Se treatments were found to cause noticeable variations in plant biomass and the concentrations of the examined elements and other quality parameters. Both Se levels promoted S accumulation in S-treated plants. Although the Se treatment had the opposite effect in shoots, it had a favorable impact on minerals (apart from Mn) in roots grown under S-limiting conditions. The S and Se relationship highlighted beneficial and/or synergistic effects for Mn and Fe in edible spinach portions. Reducing sugars were synergistically boosted by adequate S and moderate Se levels in roots, while in shoots, they were accumulated under moderate-or-higher Se and excessive S. Furthermore, the concentration of the quantified organic acids under S-deficient conditions was aided by various Se levels. In roots, moderate Se under high S application enhanced both malic acid and citric acid, while in the edible parts, higher Se under both adequate and elevated S levels were found to be advantageous in malic acid accumulation. Moreover, by elevating S levels in plant tissues, total protein concentration increased, whereas both moderate and high Se levels (Se1 and Se2) did not alter total protein accumulation in high S-applied roots and shoots. Our findings show that the high S and medium Se dose together benefit nutrient uptake; additionally, their combinations support soluble sugars and organic acids accumulation, contributing ultimately to the nutritional quality of spinach plants. Moreover, consuming 100 g of fresh red spinach shoot enriched with different Se and S levels can contribute to humans’ daily micronutrients intake.

## 1. Introduction

Sulfur (S) is a versatile macronutrient for the synthesis of many important physiological and cellular metabolites of plants, and is needed against various abiotic and biotic stresses [1,2]. A balanced and adequate S level has a vital role in producing high-quality vegetables and cereals due to the increased demand for nutritious food and diets [3]. The interactive effect of S with macronutrients, including N, P, and K, has been extensively studied [4,5]. Additionally, S exerts a role in maintaining homeostasis of certain essential micronutrients, such as molybdenum (Mo), manganese (Mn), zinc (Zn), iron (Fe), and copper (Cu), which play a crucial role in metabolic processes in plants as enzyme cofactors and components of chlorophyll and cell membrane molecules, in addition to their essentiality for protein synthesis, photosynthesis, respiration, DNA synthesis, and the electron transport chain [6,7]. S interaction with other nutrients, either macro- or micronutrients, tends to enhance or affect the growth of crops, and, subsequently, their production, yield, and quality, by influencing the uptake and utilization of essential nutrients [8]. Additionally, S can improve soluble sugars, protein content, and kernel taste by increasing plants’ flavor and sweetness [8]. Selenium (Se) is a trace element considered beneficial for plants but essential for the health of humans and animals. It has received considerable attention as it possesses various properties such as antioxidative, anticancer, anti-inflammatory, and antimicrobial, and also has antiviral properties [9]. Recently, it has been reported that for COVID-19 survival, patients having recommended Se levels in their bodies meant that the virus did not affect them severely compared to other patients [10]. Se can be crucial in reducing oxidative stress, stimulating thyroid hormone, regulating the immunological system, and enhancing human reproductive status [11]. Additionally, Se showed beneficial effects for plant growth and quality, but its essentiality for plants has yet to be established. For instance, Se treatment with sodium selenate at 3 mg L^−1^ in tomatoes boosted shoots and roots biomass, while at 10 mg L^−1^, it slightly decreased biomass accumulation. Compared to the control, fruit output was increased by 25.3% when foliar sodium selenate was applied at a lower dose of 3 mg L^−1^ [12]. Moreover, a previous study demonstrated that low Se levels improved the quality of potatoes by reducing potato tuber discoloration [13]. It has been reported that Se has enhanced mineral content and antioxidant capacity in various Se-enriched foods [14]. Se exhibited effectual changes in the uptake and distribution of mineral nutrients and some physiological aspects in rape and wheat seedlings [15]. An early study indicated that Se application increased the concentration of microelements such as Fe [16]. Moreover, a contemporary report indicated that Se application enhanced the Mn, Zn, and Fe concentrations in tomatoes, thus improving plant micronutrients accumulation [17]. It is worth noting that food quality is determined by essential factors such as organic acids and soluble sugars [18]. Subsequently, it has been well documented that Se elevates the content of soluble sugar such as glucose and fructose, amino acids, and other bioactive compounds, including vitamins. Se also affects the proteins involved during the metabolism of carbohydrates, amino acids, and secondary metabolism, ultimately improving the nutritional quality of plants [19]. Various reasons contribute to Se restriction in soil, including that most farmers do not apply Se fertilizers to agricultural land. This might be attributed to their knowledge related to the price of Se and may be a frequent negative impact of Se on yield if it is applied at a high level. Additionally, most farmers may not be aware of Se’s beneficial effects on plant development and its essentiality for human and animal health. Furthermore, crop production makes S fertilization necessary; therefore, S is often part of fertilizers. Se has chemical and physical similarities to S, which can be taken up by the roots via the S transporters and assimilated via the same S assimilatory pathway. Moreover, the plant’s S transporters can uptake S rather than Se via a low-affinity transport system, which might ultimately affect Se absorption through roots in the presence of S in soil. Accordingly, the foliar application of Se might be a good strategy for nutrient uptake, improving plant quality and nutritional values by providing a precise evaluation of plant performance grown under both Se and S enrichment. Spinach is a great nutrient-rich vegetable as it is a good source of minerals, especially calcium (Ca), potassium (K), magnesium (Mg), and Fe, in addition to vitamins C, B6, and B9, carotenoids, and several bioactive compounds [20]. Hence, the combined effect of S and Se on micronutrient uptake and pattern is not much studied; the authors carried out the current report to follow up the conceptual framework of the Se and S interactive effect. Accordingly, the authors intended to test the hypothesis that under S and Se enrichment, micronutrient uptake will be positively affected in both the roots and shoots of spinach plants.

## 2. Results

### 2.1. Spinach Plants Biomass

Visual differences among varied S treatments in spinach plants were observed, as shown in Figure 1. Plant leaves were small and had a pale green to yellowish–green color in S-deficient plants, whereas plants supplied with adequate and high S levels exhibited a better performance (Figure 1). Results showed significant differences regarding the biomass parameter, i.e., dry matter (DM), in both roots and shoots. In Se-less treated plants and adequate S (S1) or high S (S2) supply, the shoot DM increased 2- and 2.7-fold, respectively, in comparison to S-deficient plants (Table 1). Results indicated that moderate Se increased shoot DM in S-deficient (S0) plants. In roots, a notable significant (*p* ≤ 0.05) decline was observed in DM with high Se (Se2) under all S levels (Table 1).

### 2.2. Sulfur and Selenium Accumulation in Plants

The availability of mineral nutrients, including S, has a greater contribution to the growth and development of plants. Se and S applications in spinach may affect nutrient uptake and distribution in plants. S accumulation in mg g^−1^ DM was determined under different levels of Se (Se0: 0 µM, Se1: 0.5 µM, and Se2: 2 µM) and S (S0: 0 mM, S1: 1 mM, and S2: 5 mM) in spinach (Figure 2). Increasing S levels enhanced S accumulation linearly in both roots and shoots. In roots, varied Se levels did not reduce S accumulation. Only a high Se level (Se2) reduced shoot S concentration under S-starved plants, where a significant (*p* ≤ 0.05) reduction of 51.7% was observed. However, under S-sufficiency and elevated levels (S1 and S2), a synergistic relationship between S and Se was obtained where both moderate (Se1) and high (Se2) Se levels, did not decline or change S accumulation, as shown in Figure 2. Regarding Se concentration, a linear increase in roots and shoots’ Se accumulation was observed with increased Se levels under both S1 and S2 conditions. Results indicated that Se accumulated more under S deprivation (S0) in both roots and shoots compared to S-supplied plants, as shown in Figure 3. A high level of S (S2) significantly reduced (*p* ≤ 0.05) Se accumulation when a high level of Se (Se2) was applied in comparison to the Se level under S-sufficiency. Hence, S and Se showed a positive interaction regarding Se accumulation in both roots and shoots under moderate Se (Se1) supply. However, an antagonistic interaction was observed under elevated S and Se (S2 and Se2) supply.

### 2.3. Micronutrients Accumulation in Spinach

Mineral nutrients-enriched foods can play vital roles in human and animal systems. Micronutrients (µg g^−1^ DM) were determined in the roots and shoots of spinach plants grown under Se and S varied treatments, as presented in Figure 4. The results showed that the absorption of all micronutrients in roots and shoots changed significantly when S and Se fertilization was applied. Among the micronutrients, the uptake of Mo was drastically inhibited by S fertilization, as high Mo concentrations were recorded in S-starved plants. Under S-deficient and higher Se (Se2) conditions, Mo concentration was increased in roots, while both moderate and elevated Se levels decreased Mo accumulations significantly (*p* ≤ 0.05) in shoots. Furthermore, a positive interaction between Se and S was found in Mo accumulation, where both moderate and high Se did not impact the Mo level by increasing the S level to S2, as shown in Figure 4A. The application of S had an impact on spinach Zn concentration. Both moderate and high S levels reduced the accumulation of Zn in roots and shoots compared to S-deprived plants (Figure 4B). Although moderate Se levels lowered Zn concentration in shoots under S-starved (S0) conditions, high Se levels increased Zn accumulation in roots. Moreover, a moderate Se under an S-adequate level (S1) significantly (*p* ≤ 0.05) elevated Zn concentration in both roots and shoots. Interaction between Se and S in relation to Zn concentration was favorable in plants when treated with moderate Se (Se1) and adequate S (S1) levels. Se displayed antagonistic behavior against Mn in the S-unfertilized control, considerably reducing (*p* ≤ 0.05) its concentration at an elevated Se level in both roots and shoots (Figure 4C). Additionally, modest Se (Se1) fertilization improved Mn concentration in both S-supplied roots. However, excessive S (S2) fertilization decreased Mn concentration compared to adequately applied S (S1) roots. As demonstrated in Figure 4C, a high Se treatment (Se2) in comparison to a moderate Se (Se1) level increased Mn accumulation in shoots fertilized with both adequate (S1) and sufficient (S2) S. It was concerning that Fe accumulation in spinach shoots had an opposite trend when compared to roots, which showed higher Fe concentration under S-deficiency when exposed to both Se-supplied levels. The application of excessive S (S2) decreased Fe accumulation in roots but increased the Fe level in shoots compared to the S-unfertilized control (S0Se0) (Figure 4D). Both adequate and higher S fertilization levels under elevated Se2 (S1Se2) and moderate Se1 (S2Se1) supported shoots’ Fe concentration significantly (*p* ≤ 0.05) compared to the S-deficient plants. Apart from Fe accumulation, Cu concentration was enhanced in roots compared to shoots in response to Se treatments under S-untreated conditions. Cu uptake in roots and shoots was shown to be positively influenced by the interaction between Se and S (Figure 4E). According to the findings, an adequate or high S level and foliar-applied Se levels acted favorably toward the absorption of micronutrients, especially Mn and Fe, in spinach shoots, as shown in Figure 4.

### 2.4. Quantification of Organic Acids and Water-Soluble Sugars

Organic acids are one of the primary taste substances in food crops [21]. Sugars represent a quality factor, especially in food ripening and flavor. Organic acids and sugars (mg g^−1^ DM) were determined in the roots and shoots of spinach under Se, and S varied treatments as listed in Table 1. In adequate S (S1) fertilized shoots, moderate Se (Se1) significantly decreased the concentration of malic acid. As opposed to a high Se (Se2) treatment, moderate Se (Se1) supported (*p* ≤ 0.05) malic acid accumulation in high S-treated shoots. Under S-deficit conditions, Se fertilization in both roots and shoots did not considerably affect oxalic acid levels. Subsequently, adequate S (S1) fertilization induced drastic levels of oxalic acid, especially in shoots. Although moderate Se significantly decreased oxalic acid accumulation under adequate S in shoots compared to Se0S1, it did not affect the oxalic acid level in roots under the same conditions (Table 1). A favorable association between S and Se was found regarding oxalic acid concentration in shoots, where it increased under higher doses of Se and S. With increasing levels of Se, plants grown under S-limiting conditions showed a linear increase in citric acid concentration, especially in shoots. Furthermore, citric acid levels in shoots remained unaltered in response to moderate Se doses under S-adequate or higher treatments, compared to S1Se0 and S2Se0, respectively. However, moderate Se levels decreased citric acid accumulation in roots under S-sufficiency (Table 1). Apart from organic acids, water-soluble sugars in spinach were quantified under Se and S enrichment. Under S-deficient conditions, moderate Se (Se1) considerably increased glucose and fructose levels in shoots compared to the control S0Se0. Glucose and fructose accumulation in either roots or shoots were affected identically under adequate S and higher Se, as both sugars were significantly (*p* ≤ 0.05) enhanced compared to the control (S0Se0). Both sugars were dramatically enhanced in the plant’s roots when S was applied at a higher concentration (S2) and under moderate or higher Se levels, compared to the control S0Se0 (Table 1), whereas glucose significantly (*p* ≤ 0.05) decreased in shoots under elevated S compared to the control S0Se0. The moderate or high Se did not affect glucose or fructose in shoots under elevated S (S2). The production of glucose and fructose in roots was significantly favored by S and Se interaction, showing an intriguing synergism. Moreover, increasing Se levels influenced sucrose concentration in shoots under S-limiting conditions. Both Se doses did not affect sucrose levels in roots grown under S-deficient conditions. Additionally, different S treatments did not impact the sucrose accumulation in shoots. Moreover, moderate Se (Se1) reduced sucrose levels in roots grown under adequate S (S1) conditions but did not alter sucrose accumulation under higher S than the control S0Se0. 

### 2.5. Quantification of Total Protein

Due to its essentiality, S is the major contributor to protein biosynthesis. On the other hand, Se could replace S in S-containing amino acids in proteins. Total protein concentration in µg g^−1^ DM was determined in spinach treated with different levels of Se (Se0: 0 µM, Se1: 0.5 µM, and Se2: 2 µM) and S (S0: 0 mM, S1: 1 mM, and S2: 5 mM) (Figure 5). S fertilization enhanced protein concentration in both roots and shoots. Under S deficiency, only a high Se level (Se2) decreased the protein level in plant shoots compared to the control (Se0S0). However, in roots, Se at a high level (Se2) did not decline the protein level compared to the control; somewhat, Se as a moderate application enhanced protein under such conditions. Increasing the S level from adequate to high showed a significant (*p* ≤ 0.05) increase in root protein concentration in plants treated with both moderate (Se1) and high (Se2) Se levels, by 2% and 3.1%, respectively (Figure 5). Both moderate or high Se levels (Se1 and Se2) did not change total protein concentrations in response to adequate or high S treatment in both roots and shoots.

## 3. Discussion

### 3.1. Se/S-Interaction

Spinach is a rich source of vitamins and minerals, including iron, Se, and other nutrients. It can accumulate much Se, primarily in its shoot and root tissues [22]. Most studies have noted various positive benefits of Se on plant growth and performance [23,24,25]. Growing plants supplemented with Se may be a more efficient way to produce Se-enriched foods to boost human health. The interactive effect of S and S might affect the uptake of certain nutrients and spinach growth and quality. Accordingly, the present study investigated the impact of S and Se interaction, specifically in micronutrient accumulation in addition to organic acids, soluble sugars, and total protein concentrations. Our results show that S deprivation resulted in remarkable (*p* ≤ 0.05) reductions in plant biomass. This can be attributed to the S role in plant growth and metabolism, where growth and yield are negatively affected by S deficiency [26]. Both roots and shoots’ DM were significantly (*p* ≤ 0.05) enhanced under S-sufficient and excessive conditions. On the other hand, Se at moderate doses under adequate S did not affect roots’ DM, compared to S1Se0. However, in shoots, an interesting positive relationship between S and Se was found due to the Se foliar application, where shoot DM was significantly (*p* ≤ 0.05) enhanced by moderate Se under sufficient S treatment compared to S1Se0. Additionally, applying moderate Se to shoots grown in the high-S medium did not alter shoot DM compared to S2Se0, therefore indicating a beneficial role of Se in increasing the biomass of the edible portion of spinach [27]. Regarding S accumulation in spinach, enhancing the sulfate pool driven by S fertilization increased S accumulation in spinach. High Se absorption in spinach roots and shoots lacking S supply have been detected. This could be attributed to the lack of competition for selenate [28] and the exposure of shoots to Se due to a foliar application that readily enters leaves tissues via mesophyll cells, most likely through S transporters [29]. Accordingly, the significantly greater buildup of Se under S-deficient conditions led to the high expression of sulfate transporters [30], due to the Se stimulation as an S analog. A synergistic relationship between S and Se relates to S concentration in spinach parts. Se at both supplied S levels did not decline or alter S accumulation in plants, indicating that selenate stimulates sulfate uptake, potentially by delaying the reduction in sulfate transporters’ abundance and/or activity [31]. Another reason could be that an increased Se supply induces the development of the group 1 isoform, Sultr1; 1, and group 2 isoform, Sultr2; 1, which promotes S accumulation in plants [32,33]. Due to its crucial role in maintaining human and animal health, Se can be administered topically to plants in smaller amounts. Se-enriched food crops can be a dietary source of Se in humans [34]. Our results indicated that Se uptake was inhibited (*p* ≤ 0.05) in both shoots and roots by the higher S application (Figure 3), which may be attributed to the antagonistic effect of a competitive ion. S and Se compete with each other because of the similarities they possess in many aspects. Results demonstrated by Liu et al. [35] align with our findings that sulfate application dramatically lowers Se concentrations in plants treated with selenite or selenate. Additionally, foliar application of varied Se levels substantially increased the accumulation of Se in both roots and aerial parts. Interestingly, despite its inhibition in S-supplied plants, Se accumulation in plants treated with moderate Se was not affected by further increasing the S level from sufficient (S1) to high (S2), showing a synergistic interrelationship between the two elements. This seems to suggest that applying S at a higher level may not have a significant impact on plants; instead, it may be necessary to supplement foods with Se at an adequate level to protect humans and animals who consume Se-enrich foods from Se toxicity. An increase in S application counteracted Se toxicity, by preventing non-specific Se incorporation into proteins, and modulating the catalytic activities of the redox enzyme [36]. 

Concerning Se accumulation in spinach edible parts (shoots), Se foliar application ensures Se accumulation in spinach plants and can secure the recommended daily dose of 60 and 70 µg Se day^−1^ for adult women and men, respectively [37]. Spinach contains approximately 91% water [38], with the results showing that consuming 100 g of fresh red spinach shoot supplemented with lower Se (Se1) and adequate S (S1) or higher S (S2) (Se1S1 and Se1S2) levels can contribute 79.2 and 77.4 µg Se day^−1^. 

Furthermore, consuming 28–35.8 g fresh weight red spinach grown under higher Se2S1 and Se2S2 treatments, respectively, can contribute to the Se dietary requirement dose of 60 µg Se day^−1^.

### 3.2. Role of Se on Micronutrient Accumulation

Se metabolically interacts with some essential elements and toxic heavy metals, with the outcome of the biochemical interactions either synergistic or antagonistic [39,40]. It is considered that the optimum Se level inhibits the effect of some heavy metals, by decreasing their uptake and translocation to the upper plant parts, which is one of the important stress tolerance mechanisms [41,42]. Our results on the micronutrient accumulation in spinach plants revealed that applying Se under S-deficiency had a varied effect on ions in roots and shoots. Mo, Zn, Fe, and Cu significantly (*p* ≤ 0.05) increased in roots and decreased in shoots under higher Se doses. In this regard, Se interactions are mostly driven by Se concentrations, and competition with these nutrients. In contrast, Se reacted differently and displayed some fascinating interaction with S in the ion–particle relationship in spinach’s different sections in response to varied S levels. In terms of micronutrients, S administration greatly (*p* ≤ 0.05) inhibited Mo accumulation, whereas, under S deficit conditions, plants rapidly absorbed it. S is known to be a potent inhibitor of Mo uptake [30], as S transporters are important candidates for low-affinity Mo uptake. Subsequently, a great reduction in Mo levels can be driven by enhancing S treatments. Previous studies suggest that the sulfate transporters and the molybdate-specific transporters of the vascular plant *Arabidopsis thaliana* belong to the same transporter family [43]. Alhendawi et al. claimed that increased Mo absorption is encouraged by a sulfate deficiency and that molybdate and sulfate compete for the same transport and carrier sites in the uptake [44]. A high level of Se increased Mo in roots, whereas Mo accumulation was suppressed in shoots following the application of Se [45]. This could result from the foliar treatment, where Se levels opposed Mo at the applied sites in shoots under deficient S conditions because of its similarity to S; Se mimics S in its deficiency, thus affecting Mo concentration by competing for the common binding sites in shoots. Besides Mo, Zn was also higher under S deprivation, regardless of changes in growth. In our study, plant uptake of Zn was impacted (*p* ≤ 0.05) by S nutrition. The antagonism with S may be responsible for reducing Zn accumulation [46]. According to a previous study [47], Zn concentrations tended to decline when S increased. According to Singh et al. [48], who studied the “effect of S on productivity, economics, and nutrient uptake in spinach”, the uptake of N, P, K, and S by the spinach crop increased with S treatment; however, Zn uptake reduced at higher S levels. It has been indicated in several trials that S lowered Zn in spinach, indicating that S has an antagonistic influence on Zn in plants [49,50]. In the absence of S, Se’s mimicking behavior influenced the Zn concentration in shoots. Zn levels were lower (*p* ≤ 0.05) in Se-treated shoots under the S deficiency condition compared to the control S0Se0. Longchamp et al. reported that high Se impeded the transfer of Zn from the root to the shoot by detecting a decrease in Zn concentration in leaves and stems with high Se compared to control plants [51]. Similarly, Se application as a moderate and high supply lowered Zn content compared to the control in strawberry tops and leaves [52]. Both S and Se demonstrated a striking antagonistic effect with Mn, but their combined adverse effects were not observed in the edible part of spinach. In a contemporary study, it was discovered that adding Se up to 5 mg L^−1^ in nutrient solution culture, as opposed to soil culture, reduced Mn uptake by the roots of Chinese brake fern. Likewise, the plant’s Mn concentration tended to drop as Se doses increased [53,54]. Notably, another study also reported a potent antagonistic interaction between Mn and Se (selenite form) [55]. Additionally, microelements, including Fe, Mn, Zn, and Cu, typically have lower quantities under greater Se treatment, especially in the shoots. Consequently, in aboveground plant parts, it has been noticed that Se enrichment via foliar application affects micronutrient uptake. This can be explained by the antagonistic behavior of Se toward Mn in S-deficient plants and the favorable behavior of Se toward Mn in shoots when S was added to the medium. Likewise, it was reported that the Mn content of wheat (*Triticum aestivum* L. cv. Manu) roots and flag leaves decreased when exposed to greater doses of Se (15 μm) in comparison to a lower application rate (5 μm Se) [56]. In the presence of S, Se foliar application may sustain the Mn accumulation in the spinach’s edible sections [57]. It has been noted that at high Se doses, Mn concentrations decreased in all plant organs, which might be attributed to the suppression of Mn uptake by roots. The accumulation of several essential microelements, including Mn, has already been decreasing [58]; accordingly, this decline may be caused by Se, which alters the plasma membrane permeability coefficient for ions. However, only a few studies have documented the mineral composition of crop plants growing in the presence of exogenous Se under varied S levels.

An alteration in microelements driven by Se levels has been observed under S-limiting conditions, but the combination of Se and S treatments showed distinct behaviors that were supportive for the majority of the minerals, particularly in shoots. In our investigation, exposure to Se treatments resulted in stimulation or inhibition of Fe accumulation. Se concentrations boosted Fe in the roots of S-deficient plants while lowering it in their shoots. However, rising Se concentration with increasing S levels synergistically improved the Fe accumulation in the edible section of the spinach, especially under adequate or excessive S and higher and lower Se, respectively (S1Se2 and S2Se1) (Figure 4D). Results indicated by Se and S interaction in shoots are of significant interest because of the leafy vegetable spinach, a rich source of minerals like Fe. Due to Fe’s importance for human health, this approach might cover the Fe dietary requirements with a greater amount. It was previously determined that Se is frequently involved in antagonistic interactions with various heavy metals, primarily Mn, Zn, Cu, Fe, and Cd and that this decreases their absorption by raising Se levels [39]. Moreover, S nutrition could be associated with the uptake of Fe in shoots, as in S-supplied shoots, Fe was abundantly absorbed compared to Fe accumulation in shoots under S-limiting conditions. This is attributable to the fact that supplying S may result in an enhancement in Fe utilization efficiency in crops [59]. Moreover, Fe was detected in a previous study under S deprivation, presumably indicating a deteriorating sink strength of the shoot for Fe [60]. Other elements can influence the uptake, accumulation, and utilization of certain elements by plants; this was the case regarding the uptake of Fe under S-sufficiency since S shortage hinders Fe acquisition and accumulation [61]. Our findings on Fe and Mn are consistent with those of Longchamp et al. [47], who demonstrated that these elements followed a similar pattern in the availability of S in belowground parts since their concentrations were enhanced at low S. However, the highest Mn level in roots was detected in response to adequate S and moderate Se. In a similar way to other cations, Se likewise significantly increased Cu in plant roots in the absence of S but noticeably decreased it in plant shoots. Our findings showed that excessive S fertilization decreased the accumulation of Cu in spinach shoots. The results were consistent with Jankowski et al. [62], who showed that S fertilization significantly decreased Cu concentration in mustard straw by 30%. Although both moderate and high Se levels under high S fertilization decreased Cu concentration in shoots, under adequate S conditions, high Se elevated Cu accumulation in shoots. According to the previous investigation [40], Se promoted Cu buildup. Our findings exhibit that under S deficiency, Se favored nearly all metal cations in roots but showed a negative impact on them in shoots where Se was administered directly via a foliar application under similar S-starved conditions. Se appears to have an impact on the translocation of the determined micronutrients. Therefore, further research is required for a better understanding of the physiological and molecular mechanisms in both plant sections under Se and S enrichment. When S was removed from the nutrient solution, it was found to be slightly harmful to micronutrient accumulation in shoots, while S deficiency enhanced their concentrations in roots. It is worth noting that Se toxicity may be reduced by the addition of S, which is probably accomplished by reducing the non-specific integration of Se into proteins and modifying the plant’s redox system. Furthermore, plants should not be fortified with Se under S deprivation; instead, adequate or high S doses could be applied to improve mineral accumulation in the Se-enriched food crops. In the current research experiment, significant differences showed sufficient evidence to disapprove and reject the null hypothesis and accept the alternative hypothesis that S and Se enrichment has a significant impact on micronutrient uptake in spinach plants.

Concerning micronutrient accumulation in Se- and S-enriched spinach shoots, varied treatments of Se and S ensure micronutrient accumulation in the edible part and can contribute to the daily dietary recommended dose of Fe, Zn, Mn (in mg), Cu, and Mo (in µg) per day for adult women and men, as listed in Table 2. According to the water content of spinach (91%), the findings demonstrated that consuming 100 g of fresh red spinach shoot supplemented with different Se and S levels can contribute to the micronutrient daily intake for humans (Table 3). For instance, under lower Se and higher S (Se1S2) applications, the accumulated Fe, Zn, Mn, Cu, and Mo can cover approximately 60.6/136%, 22.5/16.4%, 183%, 40.1%, and 130%, respectively, of female–male daily intake. 

Moreover, Mn accumulation can successfully cover the recommended dose for females/males by consuming 51.4/65.7, 46/59, 54.5/69.6, and 63.8/50 g fresh red spinach shoots grown under Se1S1, Se2S1, Se1S2, and Se2S2 treatments, respectively, while the intake of 92.5, 84.7, 77, and 77 g of fresh spinach shoots grown under similar conditions can cover the recommended dose for Mo.

### 3.3. Importance of Se for Primary Metabolites and Food Quality

Organic acids and water-soluble sugars are significant metabolites that directly reflect the sensory properties of fruits and vegetables, including taste and flavor [3]. It has been discovered that soluble sugar and soluble protein could be considerably regulated and increased in plants by Se [64,65]. Sucrose, glucose, and fructose are known as essential photosynthates and are the basis for all food energy, while organic acids such as citrate, oxalate, and malate are produced as intermediates in the citric acid cycle and have a crucial role in metabolism [66]. According to the findings of the present investigation, roots’ glucose, and fructose levels were significantly increased by moderate or higher Se at high S levels. Similarly, high Se influences sucrose and the reducing sugars accumulation at moderate S levels in shoots.

An intriguing interaction between S and Se in roots could be attributed to the fact that S has direct consequences for sugar accumulation [64], while Se acts as a beneficial trace element of favorable effects [67]. Furthermore, it was observed in the combination treatments, where moderate Se showed synergy in sugars’ accumulation in shoots under high S levels but produced an adverse effect when Se was applied at high levels, especially glucose concentration. To maintain proper sugar levels in plants, the optimum S and Se treatments may be crucial. It is most likely that the drop in glucose accumulation in high S-supplied shoots treated with high Se might be a sign of the reduction of the photosynthetic rate. Our findings agreed with a previous report [46], which found that Se and S had a synergistic impact on glucose, fructose, and sucrose levels. It has also been discovered that Se could influence the activities of acid invertase (AI), neutral invertase (NI), and sucrose synthase (SS) [68]. Additionally, the buildup of soluble sugars under Se treatments is significantly influenced by AI and NI. The breakdown of sucrose is aided by increased AI and NI activity, which also raises levels of glucose and fructose [69]. Foliar application of silicon (Si) and Se improved the growth, yield, and quality characteristics of cucumber under field conditions; moreover, Se application boosted the accumulation of soluble sugars in cucumber [70]. Lidon et al. [64] reported that the addition of selenite (Na_2_SeO_3_) and selenate (Na_2_SeO_4_) enhanced the amount of sugar (including sucrose, glucose, and fructose) in rice grains. Se treatment increased the levels of soluble carbohydrates (glucose and fructose) in tomatoes [19]. Concerning organic acids accumulation, our findings showed that in S-deficient plants, Se treatments enhanced organic acids greatly by increasing citric acid and malic acid in their shoots, but oxalic acid concentration remained unchanged. Our findings agreed with Hu et al. [70], who demonstrated that Se application increased the levels of citric acid, malic acid, and oxalic acid. Nevertheless, S and Se interaction positively impacted organic acids’ concentrations under moderate or higher Se and high S-supplied shoots. Organic acids also reflect a plant’s metabolic condition and capacity for survival by upholding its basal metabolism [71], and therefore, they aid in developing disease resistance [72]. For instance, *Fusarium wilt* in faba bean is significantly inhibited by tartaric acid and malic acid [73]. It is worth noting that nano Se treatments can enhance pathogen resistance in plants [74]; moreover, foliar-spraying nano Se increased organic acids, mainly lactic acid, malic acid, and citric acid. Accordingly, it may be concluded that Se biofortification could trigger plants to develop a mechanistically important defense system against pathogens and improve the growth and quality of food plants and their other nutritional characteristics.

According to our findings, elevated Se in S-deficient (S0) shoots resulted in a decrease in total protein levels in spinach. When Se disrupts normal S metabolism by replacing S with Se in S-containing amino acids, the tertiary structure of a protein can be altered. The production of Se toxicity is driven by these catalytic alterations [75]. Additionally, it may be generated by elevated Se application under S-starved circumstances. The amount of total protein in the spinach grew progressively when the S supply increased. The findings of this study are supported by [48], who noted that the application of S considerably boosted the protein output of spinach crops. Because S is crucial for protein formation, S-deprivation lowers the synthesis of S-containing amino acids and consequently inhibits the synthesis of proteins [76]. Spinach’s total protein concentration improved as a result of S supply. When S was present in the nutrient medium, the application of Se, regardless of its concentrations, did not markedly affect total protein accumulation. Se had a more distinct beneficial impact when combined with S. In our study, high S fertilization positively impacted total protein accumulation at both Se levels. Applying high Se under elevated S supply did not diminish total protein concentration. This can be attributed to moderate or high Se application being too low to alter total protein levels. The Se dose is essential in influencing total protein concentration [77]. However, it was interesting to observe that the interaction between Se and S improved the protein synthesis of spinach. Moreover, Se application, combined with S, improves the nutritional and biochemical quality of food crops.

## 4. Materials and Methods

### 4.1. Experimental Design and Plant Material

The experiment was conducted in a controlled growth chamber at the Institute of Plant Nutrition and Soil Science, Kiel University, Kiel, Germany. Spinach (*S. oleracea*) (cultivar spinach Reddy F1) was selected for this study. Seeds were soaked in 2 mM calcium sulfate solution for 10–12 days for better germination. Seedlings were transferred into 5 L black plastic pots in hydroponic conditions. The pots were kept under climatic chamber conditions at 17 °C (day) and 13 °C (night) with a 14 h photoperiod. The relative humidity was kept at 50%.

### 4.2. Treatment Combinations and Chemical Characteristics

Three levels of S (S0 = 0 mM, S1 = 1 mM, and S2 = 5 mM K_2_SO_4_) and Se (Se0 = 0 µM, Se1 = 0.5 µM, and Se2 = 2 µM, Na_2_SeO_4_) were used to investigate the influence of varied S and Se treatments on micronutrients accumulation, in addition to soluble sugars, organic acids, and protein concentrations. Nine different treatment combinations were used, as follows: (1) S0Se0 (control); (2) S0Se1; (3) S0Se2; (4) S1Se0; (5) S1Se1; (6) S1Se2; (7) S2Se0; (8) S2Se1; and (9) S2Se2. Pots were organized in a completely randomized design (CRD) and replicated four times. Chemical characteristics of the basic solutions for all pots consisted of macronutrients including Ca(NO_3_) = 2 mM; NH_4_H_2_PO_4_ = 0.5 mM; MgCl_2_ = 0.5 mM; KNO_3_ = 2 mM; and micronutrients such as H_3_BO_3_ = 10 µM; MnSO_4_ = 2 µM; ZnSO_4_ = 0.5 µM; CuSO_4_ = 0.3 µM; (NH_4_)_6_Mo_7_O_24_ = 0.01 µM; and Fe-EDTA = 200 µM. The source of S was K_2_SO_4_ and it was applied via root medium, whereas the Se source was Na_2_SeO_4_ and it was applied foliarly with the help of a brush. The pH for all pots was kept at approximately 6.5. A chemical “Silwet” was used as a wetting agent (0.1 µM/50 mL of solution) for easy absorbance of Se by plant tissues. The nutrient solution was replaced twice a week. Se was applied four times to the leaves after one month of seedlings were transferred till harvesting. There were two plants per pot; subsequently, each time, 2.5 mL of sodium selenate from the stock Se solution was applied to the specific plant that had been exposed to 0.5 µM Se. Likewise, 10 mL of sodium selenate from the Se stock solution was applied to each individual plant that received 2 µM Se treatment. The growing period was kept for 45 days. After harvesting, roots and shoots were cleaned with deionized water and dried. Samples were frozen in liquid nitrogen and kept at −20 °C. Roots and shoots were dried at −55 °C for 2–3 days in a freeze dryer (Gamma1–20, Christ Osterode am Harz, Germany). After drying, the DM was recorded, and the samples were ground finely using a grinder machine (FOSS Cyclotec^TM^ 1093) and kept for further analysis.

### 4.3. The Null Hypothesis and an Alternative Hypothesis

The authors wish to determine that Se and S enrichment significantly affects micronutrient uptake. Accordingly, in our setting, the experiment was proposed to document evidence from the perspective of the null hypothesis (H_0_), which indicates no relationship between Se and S applications and micronutrient uptake. Additionally, the experiment also follows a similar valued principle from the perspective of the alternative hypothesis (H_1_), which states that Se and S enrichment positively affect micronutrient uptake.

### 4.4. Mineral Analyses Using ICP-MS

The concentrations of Se and micronutrients such as Zn, Mo, Fe, and Cu were determined by using inductively coupled plasma mass spectrometry (ICP-MS, Agilent Technologies 7700 Series, Böblingen, Germany) according to the method described by Jezek et al. [78].

### 4.5. Sulfur Determination

The concentration of S in plant samples was determined using an elemental analyzer (Flash EA1112, Thermo Fisher Scientific, Milano, Italy). Samples were prepared and measured according to the methodology described by Zörb et al. [79].

### 4.6. Determination of Organic Acids and Water-Soluble Sugars

Organic acids (oxalic acids, malic acids, and citric acids) and water-soluble sugars such as glucose, fructose, and sucrose in plant samples were determined by using ion-chromatography (IC-5000 Dionex/Thermo Scientific, Waltham, MA, USA), as described by Cataldi et al. [80].

### 4.7. Total Protein Quantification

For protein extraction, a method described by Damerval et al. [81] and modified by Zörb et al. [82] was used. Then, 100 mg fine ground and homogeneous powder of each sample was added to 1.6 mL ice-cold solution 1 (solution 1 consisted of 1.6 mL acetone, 160 mg TCA, and 80 µL 1 M DTT). After vortexing and sonication in ice for 15 min, samples were briefly mixed by inversion and kept in a freezer at −20 °C. After a minimum of 30–60 min, they were vortexed again and were centrifuged at 13,000 rpm for 15 min at 4 °C. Subsequently, the pellets were kept, and the supernatants were discarded. Then, 1.5 mL of solution 2 was added to the pellet, along with 6 µL 0.5 M EDTA (solution 2 consisted of 1.5 mL acetone and 75 µL 1 M DTT solution). After well vortexing and sonication in ice for 15 min, mixing was performed again by inversion before keeping it in the freezer at −20 °C. After 1 h, samples were vortexed before centrifugation at 13,000 rpm for 15 min at 4 °C, with the pellets again kept and the supernatants discarded. Then, 1.5 mL of solution 2 was added to the pellet. The samples were vortexed and then centrifuged for 15 min at 4 °C and 13,000 rpm. Additionally, supernatants were removed and pellets were dried in a vacuum centrifuge at 30 °C for 40 min and protein powder (dried pellets) was stored at −20 °C. For dissolving the pallet, 1 mL lysis buffer containing 8 M urea, 2 M thiourea, 4% CHAPS, 30 mM tris pH 8.8, and 5 µL proteinase inhibitor cocktail were added. To increase the dissolving membrane-bound protein, samples were well shaken at 33 °C for 2 h and then centrifuged. The supernatant was stored at −20 °C for further quantification. The total protein quantification (µg g^−1^ DM) in the spinach roots and shoots was determined using the Bradford method [83].

### 4.8. Statistical Analysis

Data were analyzed using a statistical program, Statistix 10, version 10.0. Two-factor ANOVA was performed, followed by multiple comparisons. Mean values were compared for all combinations of the factor levels using the LSD test at level (α = 0.05). A software program, “GraphPad” Prism, version 8.4.2., was used to create the bar graphs. Labeling the bars with small letters in the graphics showed whether the various treatment levels had significant differences.

## 5. Conclusions

Leafy vegetables like spinach might benefit from applying sulfate combined with foliar Se supply to counteract the adverse effects of a high Se concentration. As a result, understanding the interaction between Se and S in the roots and shoots’ parts under varied levels is critical for producing Se-enriched plants. The results indicated that spinach plants should not be enriched with Se under S-limiting conditions, because Se toxicity may decline due to S-adequate application. Significantly, in shoots, moderate and higher Se increased phytomass under adequate S supply. Our findings demonstrated that Se could limit micronutrient uptake in shoots, especially under S-deprivation. In contrast, Se treatments induced micronutrient accumulation at adequate or high S levels. Because spinach is a rich source of minerals, especially Fe, which is important for human health, Se and S crosstalk enhanced Fe accumulation in edible parts (shoots), especially under higher Se and S enrichment. This can be an efficient approach to cover the Fe dietary requirements outstandingly. Additionally, the dietary intake of Zn and Cu for humans has been enhanced. Moreover, Se and S enrichment is positively associated with primary metabolites (including glucose, fructose, malic, and citric acids), influencing quality attributes and consumers’ preferences. In this regard, foliar-applied Se combined with root-mediated S could efficiently create Se-rich food. Further enzymatic and molecular investigation is needed to fully understand the underlying mechanism of micronutrient uptake, acquisition, and translocation in plant parts, as influenced by S and Se crosstalk.

## Figures and Tables

**Figure 1 ijms-24-12766-f001:**
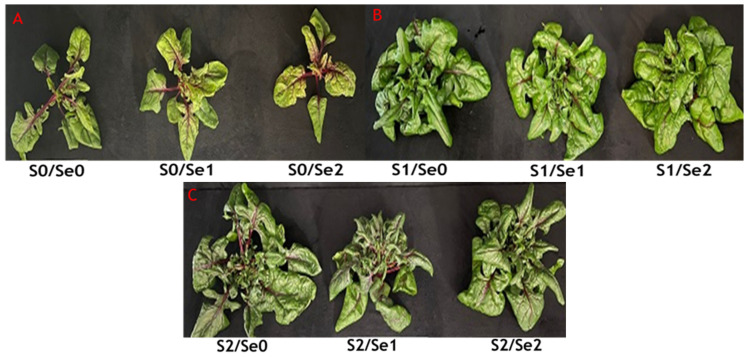
Comparison of red spinach plants (cultivar: Reddy F1) cultivated in a hydroponic system treated with S at three levels (S0: 0 mM, S1: 1 mM, and S2: 5 mM K_2_SO_4_) and Se at three levels (Se0: 0 µM, Se1: 0.5 µM, and Se2: 2 µM Na_2_SeO_4_). (**A**–**C**) Top view of the spinach plants under different S and Se treatments.

**Figure 2 ijms-24-12766-f002:**
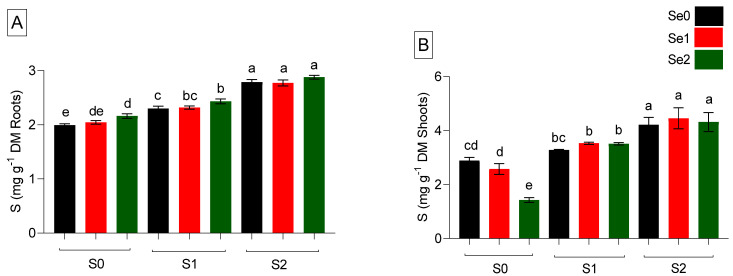
Sulfur accumulation (mg g^−1^ DM) in roots (**A**) and shoots (**B**) in spinach plants grown in a hydroponic system and treated with S at three levels (S0: 0 mM, S1: 1 mM, and S2: 5 mM K_2_SO_4_) and Se at three levels (Se0: 0 µM, Se1: 0.5 µM, and Se2: 2 µM Na_2_SeO_4_). The data presented are the means ± (SE) of four replicates. Different letters show statistically significant differences among all the treatments (*p* ≤ 0.05; LSD test).

**Figure 3 ijms-24-12766-f003:**
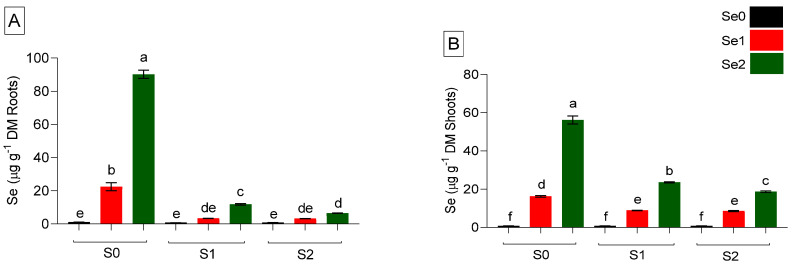
Selenium accumulation (µg g^−1^ DM) in roots (**A**) and shoots (**B**) in spinach plants grown in a hydroponic system and treated with S at three levels (S0: 0 mM, S1: 1 mM, and S2: 5 mM K_2_SO_4_) and Se at three levels (Se0: 0 µM, Se1: 0.5 µM, and Se2: 2 µM Na_2_SeO_4_). The data presented are the means ± (SE) of four replicates. Different letters show statistically significant differences among all the treatments (*p* ≤ 0.05; LSD test).

**Figure 4 ijms-24-12766-f004:**
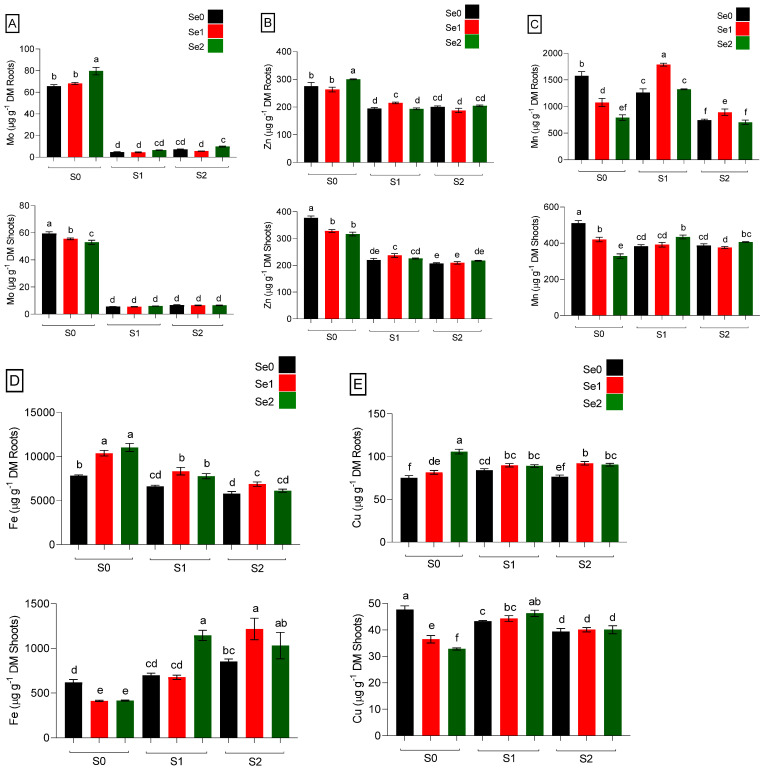
Micronutrients accumulation in roots and shoots, respectively. (**A**) Mo; (**B**) Zinc; (**C**) Mn; (**D**) Fe; and (**E**) Cu (µg g^−1^ DM) of spinach plants grown in a hydroponic system and treated with S at three levels (S0: 0 mM, S1: 1 mM, and S2: 5 mM K_2_SO_4_) and Se at three levels (Se0: 0 µM, Se1: 0.5 µM, and Se2: 2 µM Na_2_SeO_4_). The data presented are the means ± (SE) of four replicates. Different letters show statistically significant differences among all the treatments (*p* ≤ 0.05; LSD test).

**Figure 5 ijms-24-12766-f005:**
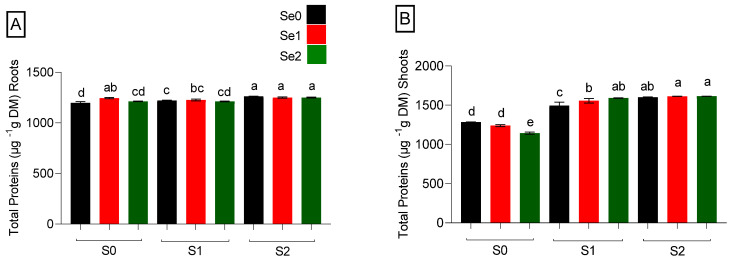
Total protein concentration (µg g^−1^ DM) in roots (**A**) and shoots (**B**) in spinach plants grown in a hydroponic system and treated with S at three levels (S0: 0 mM, S1: 1 mM, and S2: 5 mM K_2_SO_4_) and Se at three levels (Se0: 0 µM, Se1: 0.5 µM, and Se2: 2 µM Na_2_SeO_4_). The data presented are the means ± (SE) of four replicates. Different letters show statistically significant differences among all the treatments (*p* ≤ 0.05; LSD test).

**Table 1 ijms-24-12766-t001:** Plant biomass (DM (g plant^−1^)); organic acids (malate, oxalate, and citrate (mg g^−1^ DM)); and water-soluble sugars (glucose, sucrose, and fructose (mg g^−1^ DM)) accumulation in roots (left) and shoots (right) of spinach plants grown in a hydroponic system and treated with S at three levels (S0: 0 mM, S1: 1 mM, and S2: 5 mM K_2_SO_4_) and Se at three levels (Se0: 0 µM, Se1: 0.5 µM, and Se2: 2 µM Na_2_SeO_4_).

Treatments		Roots							Shoots						
		Plant Biomass (g Plant^−1^)	Organic Acids (mg g^−1^ DM)			Water-Soluble Sugars (mg g^−1^ DM)			Plant Biomass (g)	Organic Acids (mg g^−1^ DM)			Water-Soluble Sugars (mg g^−1^ DM)		
		DM	Malate	Oxalate	Citrate	Glucose	Fructose	Sucrose	DM	Malate	Oxalate	Citrate	Glucose	Fructose	Sucrose
S0	Se0	0.85 ± 0.02 ^de^	1.8 ± 0.17 ^de^	24.9 ± 0.42 ^bc^	1.8 ± 0.08 ^ef^	1.7 ± 0.37 ^d^	0.7 ± 0.08 ^d^	8.1 ± 0.13 ^ab^	2.78 ± 0.05 ^f^	7.3 ± 0.41 ^d^	115.9 ± 2.69 ^d^	2.9 ± 0.16 ^c^	7.6 ± 0.60 ^d^	2.6 ± 0.11 ^d^	7.2 ± 0.45 ^cde^
	Se1	0.95 ± 0.01 ^d^	1.5 ± 0.11 ^ef^	24.0 ± 0.58 ^bc^	2.9 ± 0.31 ^a^	2.2 ± 0.18 ^d^	1.3 ± 0.10 ^d^	8.6 ± 0.60 ^ab^	3.64 ± 0.04 ^e^	13.7 ± 1.80 ^b^	116.1 ± 0.92 ^d^	6.4 ± 0.03 ^b^	23.0 ± 1.17 ^a^	5.3 ± 0.27 ^a^	10.9 ± 0.97 ^b^
	Se2	0.36 ± 0.01 ^f^	1.2 ± 0.12 ^f^	25.7 ± 0.49 ^b^	3.0 ± 0.17 ^a^	1.3 ± 0.10 ^d^	0.7 ± 0.08 ^d^	9.4 ± 0.93 ^a^	3.03 ± 0.05 ^f^	11.7 ± 1.09 ^b^	115.1 ± 1.15 ^d^	11.6 ± 0.54 ^a^	13.2 ± 0.67 ^b^	2.9 ± 0.23 ^cd^	75.9 ± 2.80 ^a^
S1	Se0	1.51 ± 0.03 ^b^	2.6 ± 0.06 ^c^	31.5 ± 0.87 ^a^	2.2 ± 0.21 ^bcd^	1.9 ± 0.14 ^d^	1.2 ± 0.06 ^d^	7.3 ± 0.26 ^bc^	8.42 ± 0.13 ^d^	7.8 ± 1.62 ^cd^	154.7 ± 3.34 ^a^	1.1 ± 0.30 ^d^	6.3 ± 1.16 ^def^	2.9 ± 0.34 ^cd^	6.7 ± 0.56 ^de^
	Se1	1.38 ± 0.01 ^b^	2.6 ± 0.13 ^c^	29.8 ± 1.14 ^a^	1.5 ± 0.04 ^f^	1.7 ± 0.24 ^d^	0.8 ± 0.05 ^d^	3.1 ± 0.35 ^e^	9.37 ± 0.06 ^c^	2.4 ± 0.02 ^e^	131.5 ± 2.10 ^c^	1.0 ± 0.18 ^d^	7.5 ± 0.54 ^de^	3.6 ± 0.13 ^b^	8.2 ± 0.45 ^bcd^
	Se2	0.67 ± 0.02 ^e^	5.0 ± 0.05 ^a^	25.7 ± 0.44 ^b^	2.6 ± 0.10 ^ab^	4.8 ± 0.41 ^c^	2.3 ± 0.19 ^c^	6.0 ± 0.54 ^cd^	9.93 ± 0.16 ^b^	13.3 ± 1.58 ^bc^	128.6 ± 1.02 ^c^	2.7 ± 0.18 ^c^	10.0 ± 0.47 ^c^	5.6 ± 0.21 ^a^	11.0 ± 0.54 ^b^
S2	Se0	1.81 ± 0.02 ^a^	2.0 ± 0.24 ^d^	23.3 ± 0.41 ^c^	2.1 ± 0.13 ^cde^	3.6 ± 0.50 ^c^	2.2 ± 0.27 ^c^	5.9 ± 0.16 ^cd^	10.37 ± 0.10 ^a^	18.1 ± 0.48 ^a^	134.5 ± 1.83 ^c^	3.0 ± 0.31 ^c^	5.4 ± 0.19 ^f^	3.3 ± 0.05 ^bc^	7.4 ± 0.22 ^cde^
	Se1	1.17 ± 0.18 ^c^	4.4 ± 0.21 ^b^	24.5 ± 0.72 ^bc^	2.4 ± 0.24 ^bc^	8.0 ± 0.71 ^b^	6.0 ± 0.46 ^b^	7.7 ± 0.26 ^b^	10.39 ± 0.17 ^a^	21.0 ± 0.88 ^a^	135.0 ± 3.40 ^c^	3.2 ± 0.32 ^c^	6.3 ± 0.28 ^def^	3.3 ± 0.09 ^bc^	10.0 ± 0.68 ^bc^
	Se2	0.45 ± 0.01 ^f^	2.7 ± 0.10 ^c^	23.2 ± 0.55 ^c^	1.9 ± 0.12 ^def^	11.8 ± 0.68 ^a^	10.4 ± 0.58 ^a^	4.9 ± 1.05 ^d^	9.72 ± 0.08 ^b^	11.1 ± 1.60 ^bc^	142.4 ± 2.86 ^b^	3.1 ± 0.23 ^c^	5.5 ± 0.22 ^ef^	3.4 ± 0.20 ^bc^	4.8 ± 0.34 ^e^

The data presented are the means ± (SE) of four replicates. Different letters in the columns show statistically significant differences among all the treatments (*p* ≤ 0.05; LSD test).

**Table 2 ijms-24-12766-t002:** Recommended daily dietary intake of micronutrients for humans [63].

Micronutrients	Male	Female
Fe	8 mg day^−1^	18 mg day^−1^
Zn	11 mg day^−1^	8 mg day^−1^
Mn	2.3 mg day^−1^	1.8 mg day^−1^
Cu	900 µg day^−1^	900 µg day^−1^
Mo	45 µg day^−1^	45 µg day^−1^

**Table 3 ijms-24-12766-t003:** Daily dietary requirements of micronutrients for humans when consuming 100 g of fresh red spinach shoots grown under different Se and S treatments.

Micronutrients	Se1S1	Se2S1	Se1S2	Se2S2
Fe	6 mg day^−1^	10.3 mg day^−1^	10.9 mg day^−1^	9.2 mg day^−1^
Zn	2.1 mg day^−1^	2 mg day^−1^	1.8 mg day^−1^	1.9 mg day^−1^
Mn	3.5 mg day^−1^	3.9 mg day^−1^	3.3 mg day^−1^	3.6 mg day^−1^
Cu	397.8 µg day^−1^	415.8 µg day^−1^	360.9 µg day^−1^	360 µg day^−1^
Mo	48.6 µg day^−1^	53.1 µg day^−1^	58.5 µg day^−1^	58.5 µg day^−1^

## Data Availability

Data are contained within the article.

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
