# Peer review of "The Interplay of Sulfur and Selenium Enabling Variations in Micronutrient Accumulation in Red Spinach"

_ijms, 2023, doi:10.3390/ijms241612766_

Round 1

Reviewer 1 Report

I ask that the comments be respected

Comments, suggestions                                                        

Line 10: Complete the information on the effect of selenium application (different doses, in different growth phases and compounds, etc.) on the yield of different plants.

I did a review (unpublished) more than five years ago and found virtually no article that claimed that Se application had a definite positive effect on the yield of cultivated plants. On the contrary, the study (90% of the articles) revealed its negative impact on the harvest. Similarly, your results indicate a negative effect of Se on phytomass formation (Table 1). The only exception is if the plant lacks sulfur and you apply little Se (tab. 1). This finding of yours is key (significant). It should also be stated in the abstract.

Line 60: Give specific literary sources, specific examples, at what doses, terms and forms of applied Se, etc., a positive effect of selenium on phytomass and on the main product (seed, fruit, etc.) was recorded.

Line 75: Justify this information. What is the cause of this knowledge? (high price of Se and frequent negative impact of Se on yield?)

Line 87: One cannot completely agree with this sentence. The effect of sulfur is relatively well studied.

Line 196 (Table 1): The observed negative effect on phytomass also affects the concentration of nutrients in the plant, so I recommend the authors to evaluate the effect of experimental variants on the total intake (absorption) of nutrients (macro and microelements) by plants in the next article. It will be a more objective indicator than the effect on nutrient concentration.

Line 260: Article no. 24 does not report knowledge of the effect of Se on phytomass, so do not report it.

Article no. 23 states in the abstract: "The tuber yield of Se-treated potato plants was higher and composed of relatively few but large tubers",  but the same author found that:

„Selenium applications of 0.01 and 0.075 mg kg-1 had no effect on the yield of immature tubers of cultivars Satu or Sini (II). However, the highest yields of cv Satu were harvested from the mature plants treated with Se at 0.075 and 0.3 mg kg-1 (II). The final number of tubers was reduced, but tuber size was increased in the Se-supplemented plants. This suggests that Se may alter the allocation pattern of assimilates". Think about it.

Line 498 -499: The amount of sulfur applied is greater than you state. You did not include sulfur from MnSO4, ZnSO4, CuSO4.

Line 500-501: Add the information in what doses (per plant or per area) and in what concentrations the selenium itself was applied:

Line 558 – 559: Why such a conclusion? After all, in only one case, the application of selenium did not reduce the formation of phytomass, and that was under the condition that sulfur was not applied. The farmer receives money for the harvest. Its aim is not to reduce the yield by applying expensive selenium. 

Author Response

Editor in Chief

IJMS MPDI

Dear Editor,

Thank you very much for editing our manuscript entitled "The interplay of Sulfur and Selenium Enabling Variations in Micronutrient Accumulation in Red Spinach" which was submitted to IJMS MPDI.

We have revised the manuscript carefully again according to the Reviewer’s comments, as you can see in “Responses to the Reviewers’ Comments”. Please find my responses to the reviewers’ comments below.

I hope that my revised manuscript is now entirely acceptable for publication in IJMS MPDI.

I look forward to receiving a favourable decision from you.

Sincerely,

Dr. Muna Ali Abdalla

Institute of Plant Nutrition and Soil Science

Kiel University

Hermann-Rodewald-Str. 2, 24118 Kiel, Germany

Email: mabdalla@ plantnutrition.uni-kiel.de

Response to the Reviewers’ Comments

Reviewer 1 (Round 1)

Comments and Suggestions for Authors

Line 10: Complete the information on the effect of selenium application (different doses, in different growth phases and compounds, etc.) on the yield of different plants.

The authors greatly thank the reviewer for his/her constructive comments and suggestions to improve our manuscript.

Response:

The sentence is modified accordingly (Lines 10-11).

I did a review (unpublished) more than five years ago and found virtually no article that claimed that Se application had a definite positive effect on the yield of cultivated plants. On the contrary, the study (90% of the articles) revealed its negative impact on the harvest. Similarly, your results indicate a negative effect of Se on phytomass formation (Table 1). The only exception is if the plant lacks sulfur and you apply little Se (tab. 1). This finding of yours is key (significant). It should also be stated in the abstract.

Response:

Many thanks for the reviewer´s comment. The finding was made in an S-deficient condition. As our main focus was on the combined effect of S and Se, several research outcomes also showed that Se, at a modest level, had a favorable impact on plant biomass. Changes have been included per the reviewer’s suggestion (Lines 17-18).

Line 60: Give specific literary sources, specific examples, at what doses, terms and forms of applied Se, etc., a positive effect of selenium on phytomass and on the main product (seed, fruit, etc.) was recorded.

Response:

Many thanks for the reviewer´s comment. Changes were included per the reviewer´s suggestion (Lines 60-63).

For instance, Se treatment with sodium selenate at 3 mg L-1 in tomatoes boosted shoot and root biomass, while at 10 mg L-1 slightly decreased biomass accumulation. Compared to the control, fruit output was increased by 25.3% when foliar sodium selenate was applied at a lower dose of 3 mg L-1.”

Reference:

Neysanian, M.; Iranbakhsh, A.; Ahmadvand, R.; Ardebili, Z.O.; Ebadi, M. Comparative efficacy of selenate and selenium nanoparticles for improving growth, productivity, fruit quality, and postharvest longevity through modifying nutrition, metabolism, and gene expression in tomato; potential benefits and risk assessment. PLOS ONE 2021, 16, e0250192.

Line 75: Justify this information. What is the cause of this knowledge? (High price of Se and frequent negative impact of Se on yield?)

Response:

Many thanks for the reviewer´s comment. Changes have been included per the reviewer´s comment (Lines 78-82).

This might be attributed to their knowledge related to the price of Se and may be a frequent negative impact of Se on yield if it’s applied at a high level. Additionally, most farmers may not be aware of Se's beneficial effects on plant development and its essentiality for human and animal health.”

Line 87: One cannot completely agree with this sentence. The effect of sulfur is relatively well studied.

Response:

Many thanks for the reviewer´s comment. Yes, The effect of sulfur is well studied, but information regarding its interactive effect with Se is scarce (especially on micronutrients). Change has been included per the reviewer´s comment (Line 92).

Line 196 (Table 1): The observed negative effect on phytomass also affects the concentration of nutrients in the plant, so I recommend the authors to evaluate the effect of experimental variants on the total intake (absorption) of nutrients (macro and microelements) by plants in the next article. It will be a more objective indicator than the effect on nutrient concentration.

Response:

Many thanks for this excellent comment. We will consider this in the following article.

Line 260: Article no. 24 does not report knowledge of the effect of Se on phytomass, so do not report it.

Response: Many thanks for the reviewer´s comment. It is now omitted from the mentioned Line 267.

Article no. 23 states in the abstract: "The tuber yield of Se-treated potato plants was higher and composed of relatively few but large tubers", but the same author found that:

Response:

Many thanks for the reviewer´s comment. Yes, you noted that the same author did not find the effect of Se on the yield of tubers (immature), but in the case of mature plants, they obtained the highest yield/increased tuber size (strict tuber number was reduced) in Se-supplemented plants. Irrespective of the Se doses/concentrations, we cited the above-mentioned researcher for this, that “the majority of studies noted various positive benefits of Se on plant growth and performance” in the discussion chapter.

„Selenium applications of 0.01 and 0.075 mg kg-1 had no effect on the yield of immature tubers of cultivars Satu or Sini (II). However, the highest yields of cv Satu were harvested from the mature plants treated with Se at 0.075 and 0.3 mg kg-1 (II). The final number of tubers was reduced, but tuber size was increased in the Se-supplemented plants. This suggests that Se may alter the allocation pattern of assimilates". Think about it.

Response:

Many thanks for the clarification.

Line 498 -499: The amount of sulfur applied is greater than you state. You did not include sulfur from MnSO4, ZnSO4, and CuSO4.

Response:

Many thanks for this critical comment. With respect to the reviewer´s comment, the salt sources for Mn, Zn, and Cu also contain S, but only in very tiny quantities (in micromolar) that we utilize to give plants these vital micronutrients. All plants, including those with low, adequate, and high S supplies, received the same amount of nutrient solutions from these micronutrient salts. S0 (0 mM) does not necessarily indicate that the plants were entirely S-deprived. But in those plants, S deficiency symptoms (including loss of growth and development appearing in very small plant biomass) were also detected even with the addition of micronutrients in S forms. Moreover, S is a macronutrient and should be applied in higher amounts (in mM and not in µM).

Line 500-501: Add the information in what doses (per plant or per area) and in what concentrations the selenium itself was applied:

Response:

Many thanks for the reviewer´s comment. Changes were included in (Lines 545-548).

There were two plants per pot; subsequently, each time, 2.5 ml of sodium selenate from the stock Se solution was applied to the specific plant that had been exposed to 0.5 µM Se. likewise, 10 ml of sodium selenate from Se stock solution was applied to each individual plant that received 2 µM Se treatment.”

Line 558 – 559: Why such a conclusion? After all, in only one case, the application of selenium did not reduce the formation of phytomass, and that was under the condition that sulfur was not applied. The farmer receives money for the harvest. Its aim is not to reduce the yield by applying expensive selenium.

Response:

Many thanks for the reviewer´s comment. The application of selenium did not reduce phytomass under S-deprivation. Similarly, in shoots, moderate and higher Se also increased phytomass under adequate S supply. And the farmers mostly prefer plant shoots as their harvest material (edible part). So in this condition, their desired yield may not be affected by applying Se at a moderate level. Changes were included in (Lines 567-568).

Reviewer 2 Report

Dear Authors,

The title of your work "The interaction of sulphur and selenium enabling changes in the accumulation of micronutrients in red spinach" seemed interesting to me and I read it with interest. However, the work has many shortcomings and errors.

Detailed notes:

1.     Abstract is too long and unstructured.

2.     The introduction rightly ends with a research hypothesis, but an alternative research hypothesis should be set against the null hypothesis, and then should be verified later in the work. However, there was no verification. This should be supplemented.

3.     The research results were discussed in detail, but not all interpretation possibilities were used

4.     The chapter "Material and methods" is generally well written, but in section 4.7. "Statistical calculations" The authors mention two times about performing a two-way ANOVA, but they do not mention what test they used to compare the means. It doesn't matter what program they used to make the drawings, it's just one of the tools authors can use.

5.     When discussing the results, the authors did not use the possibility of interpretation using letter marking, which makes it possible to compare significant differences, but also to draw attention to homogeneous results. Please pay more attention to the interpretation of test results using significance tests.

6.     Conclusions should be summarizing and generalizing. The most important conclusion from the work was that "Se can limit the uptake of microelements by shoots, especially in the conditions of S deficiency". This should have been emphasized in the abstract. One conclusion should be directed towards the future.

The work requires a moderate edition of the English language

Author Response

Reviewer 2 (Round 2)

The title of your work "The interaction of sulphur and selenium enabling changes in the accumulation of micronutrients in red spinach" seemed interesting to me and I read it with interest. However, the work has many shortcomings and errors.

The authors greatly thank the reviewer for his/her constructive comments and suggestions to improve our manuscript.

Detailed notes:

  1. Abstract is too long and unstructured.

Response:

Many thanks for the reviewer´s comment. The abstract was modified as suggested by the reviewer. Changes are highlighted in yellow color.

  1. The introduction rightly ends with a research hypothesis, but an alternative research hypothesis should be set against the null hypothesis, and then should be verified later in the work. However, there was no verification. This should be supplemented.

Response:

Many thanks for the reviewer´s comment. The null and alternative hypotheses were established in the material and methods (Lines 554-560) and verified in the discussion part (Lines 432-435).

  1. The research results were discussed in detail, but not all interpretation possibilities were used

Response:

Thank you for bringing this up. There is an abundance of literature on the effects of S and Se on plants, but less is known about the combined impacts of these two factors on micronutrient uptake, and other processes. We did our best to enrich the discussion (Lines 316-335 and 436-455). In addition to changes highlighted in yellow.

  1. The chapter "Material and methods" is generally well written, but in section 4.7. "Statistical calculations" The authors mention two times about performing a two-way ANOVA, but they do not mention what test they used to compare the means. It doesn't matter what program they used to make the drawings, it's just one of the tools authors can use.

Response:

The test type is now mentioned in section 4.7. Statistical analysis (Lines 597-599).

  1. When discussing the results, the authors did not use the possibility of interpretation using letter marking, which makes it possible to compare significant differences, but also to draw attention to homogeneous results. Please pay more attention to the interpretation of test results using significance tests.

Response:

Many thanks for the reviewer´s comment. According to the reviewer´s suggestion, changes have been performed (Lines 276, 283, 301, 338, 352 and 360).

  1. Conclusions should be summarizing and generalizing. The most important conclusion from the work was that "Se can limit the uptake of microelements by shoots, especially in the conditions of S deficiency". This should have been emphasized in the abstract. One conclusion should be directed towards the future.

Response:

Many thanks for the comment and suggestion. The important conclusion of our work is included per the reviewer´s suggestion (Lines 609-611). Additionally, one conclusion directed toward the future was included (Lines 620-622).

“Further enzymatic and molecular investigation is needed to fully understand the underlying mechanism of micronutrient uptake, acquisition, and translocation in plant parts as influenced by S and Se crosstalk.”

Reviewer 3 Report

Dear authors,

 The manuscript is  well written. I have read it with pleasure. Some suggestion for improving are below:

Lines 77, 427, 456 – unnecessary dot,

Lines 174 – 175 – Figure 4C is about Mn, not Zn. Moreover, this statement is not so obvious looking at Zn (or Mn?) concentrations. In fact Mn concentrations in S2 are higher in roots comparing to shoots, and for Zn the concentrations are similar in roots and shoots. So, please chcek this again.

Line 329 – these sentence is probably unfinished „ foliar application of….” what?

 Lines 381, 400, 465 – please check and correct

Material and methods – describe more accuratelly the way of application of Se with brush – why did you use the brush? How did you get particular (the same) amount of fertilizer on each plant?

 Two more questions:
1. What is exactly new in your research? Please underline this more clearly also in the discussion/ conclusion sections. While reading discussion section it seems that Se and S suplementation have been extensively studied so far (incuding spinach), so what distinguishes your research?

2. Could you compare concentrations of micronutrients which you got with some dietary recommendations regarding the content of micronutrients in food/consumed plants, or  (if there aren’t any) refer the found concentrations to the content of spinach most frequently reported in the literature?

English fine

Author Response

Reviewer 3 (Round 1)

Dear authors,

 The manuscript is well written. I have read it with pleasure. Some suggestion for improving are below:

Lines 77, 427, 456 – unnecessary dot,

Response: Unnecessary dots in the mentioned lines are removed.

Lines 174 – 175 – Figure 4C is about Mn, not Zn. Moreover, this statement is not so obvious looking at Zn (or Mn?) concentrations. In fact Mn concentrations in S2 are higher in roots comparing to shoots, and for Zn the concentrations are similar in roots and shoots. So, please check this again.

Response: Thanks for pointing out this. We noted this after we submitted the manuscript to the journal. It might be mistakenly done during proofreading. The statement is checked and corrected.

Line 329 – these sentence is probably unfinished „ foliar application of….” what?

Response:

Many thanks for this comment. Here, the wordings need to be arranged. The sentence is rewritten in line 346 accordingly and is highlighted in yellow.

 Lines 381, 400, 465 – please check and correct

Response: Lines 381, 400 are rechecked and corrected. The sentence is omitted in Line 465, as there is no need for such information (already mentioned at the beginning of the paragraph, Lines 496-497).

Material and methods – describe more accurately the way of application of Se with brush – why did you use the brush? How did you get particular (the same) amount of fertilizer on each plant?

Response:

An accurate amount of Se or any nutrient could be more precisely applied as foliar using a soft brush instead spraying, where there is a high chance of losing the exact amount. We found this more accessible and precise compared to other tools. We have explained in the materials and methods section that the same amount of fertilizer was applied to each plant (Lines 545-548).

There were two plants per pot; subsequently, each time, 2.5 ml of sodium selenate from the stock Se solution was applied to the specific plant that had been exposed to 0.5 µM Se. likewise, 10 ml of sodium selenate from Se stock solution was applied to each individual plant that received 2 µM Se treatment.”

 Two more questions:
1. what is exactly new in your research? Please underline this more clearly also in the discussion/ conclusion sections. While reading discussion section it seems that Se and S supplementation have been extensively studied so far (including spinach), so what distinguishes your research?

Response:

Many thanks for this critical comment. S and Se have each been investigated separately. However, there hasn't been much research done yet on the relationship between the importance of S as a macronutrient and its antagonistic influence on Se treatment. In addition, we also applied Se to the leaves of the spinach using foliar application, to see if the uptake of foliar Se was impacted by S applied through the root system or not due to potential antagonistic effects with S. Additionally, it was determined what the nutritional and quality status of the plant would be in spinach when their combination (both S and Se) was used.

As a result, this study was novel in several ways, including the way in which Se was applied to determine whether or not S and foliar Se were still antagonistic to one another and to determine how well micronutrients were absorbed, which is essential for the nutritional value and quality of edible leafy vegetables like spinach.

  1. Could you compare concentrations of micronutrients which you got with some dietar recommendations regarding the content of micronutrients in food/consumed plants, or (if there aren’t any) refer the found concentrations to the content of spinach most frequently reported in the literature?

Response:

The authors would like to immensely thank the reviewer for this excellent comment. A new table for the recommended daily dietary requirements of micronutrients for humans was included in the discussion part. Additionally, all micronutrients were calculated per 100 g spinach FW (Table 3) to ensure how much the daily recommended dose of each micronutrient can be performed under Se and S enrichment (Lines 436-453).

Round 2

Reviewer 2 Report

Daer Authors,

The authors made all changes and additions as suggested by the reviewer.

Minor English language edition required.